# Trends in COVID-19 hospital outcomes in England before and after vaccine introduction, a cohort study

Peter D. Kirwan [1] ✉, Andre Charlett[2], Paul Birrell [1,2], Suzanne Elgohari[2], Russell Hope[2], Sema Mandal[2], Daniela De Angelis[1,2,3] & Anne M. Presanis[1,3]

Widespread vaccination campaigns have changed the landscape for COVID-19, vastly altering symptoms and reducing morbidity and mortality. We estimate trends in mortality by month of admission and vaccination status among those hospitalised with COVID-19 in England between March 2020 to September 2021, controlling for demographic factors and hospital load. Among 259,727 hospitalised COVID-19 cases, 51,948 (20.0%) experienced mortality in hospital. Hospitalised fatality risk ranged from 40.3% (95% confidence interval 39.4–41.3%) in March 2020 to 8.1% (7.2–9.0%) in June 2021. Older individuals and those with multiple co-morbidities were more likely to die or else experienced longer stays prior to discharge. Compared to unvaccinated people, the hazard of hospitalised mortality was 0.71 (0.67–0.77) with a first vaccine dose, and 0.56 (0.52–0.61) with a second vaccine dose. Compared to hospital load at 0–20% of the busiest week, the hazard of hospitalised mortality during periods of peak load (90–100%), was 1.23 (1.12–1.34). The prognosis for people hospitalised with COVID-19 in England has varied substantially throughout the pandemic and according to case-mix, vaccination, and hospital load. Our estimates provide an indication for demands on hospital resources, and the relationship between hospital burden and outcomes.

It is now well established that a segment of the population who acquire severe acute respiratory syndrome coronavirus 2 (SARS-CoV-2) infection, the virus responsible for coronavirus disease 2019 (COVID-19), in the community will require hospitalisation, potential escalation to intensive care facilities, and may die in hospital or soon after discharge. COVID-19 has been shown to disproportionately impact older people and those with multiple co-morbidities, compared to younger, healthier individuals, and there is considerable evidence that these factors heavily influence prognosis following hospital admission for COVID-19[1–4].

The extensive vaccination campaign in England during 2021 has dramatically changed the outlook for COVID-19, lessening symptoms and reducing morbidity and mortality[5,6]. Despite widespread and high levels of vaccination, however, individuals continue to experience COVID-19 infection severe enough to require hospitalisation. Several studies have previously examined COVID-19 hospitalised fatality risk (HFR) in England according to baseline demographic factors[1–3,7–9] but there is little information about prognosis in the current context of vaccination across the population and how this might be related to hospital load.

We aimed to investigate trends in mortality within 90 days of hospitalisation with COVID-19 among a national cohort of all people hospitalised with community-onset COVID-19 in England and how these trends vary according to vaccination status, hospital load, and other factors. We apply statistical methods which account for competing outcomes to estimate absolute and relative risks of hospitalised fatality, and lengths of stay in hospital by outcome, and control for or assess the potential impact of different biases.

[1]Medical Research Council Biostatistics Unit, School of Clinical Medicine, University of Cambridge, Cambridge, UK. [2]UK Health Security Agency, London, UK. [3]These authors jointly supervised this work: Daniela De Angelis, Anne M. Presanis. ✉e-mail: peter.kirwan@mrc-bsu.cam.ac.uk

**Table 1 | Characteristics of the study population compared with all people hospital-onset COVID-19 in England and all people with PCR-confirmed community-acquired COVID-19 in England**

| Characteristic | Study population (hospitalised for COVID-19 in England) n (%) | All people with hospital-onset COVID-19 in England n (%) | All people with PCR-confirmed community-acquired COVID-19 in England n (%) |
|---|---|---|---|
| Total | 259,727 (100%) | 208,851 (100%) | 6,616,231 (100%) |
| **Age** | | | |
| 0–14 | 6650 (2.6%) | 2632 (1.3%) | 892,640 (13.5%) |
| 15–24 | 8972 (3.5%) | 4417 (2.1%) | 1,265,595 (19.1%) |
| 35–44 | 44,094 (17.0%) | 13728 (6.6%) | 2,242,751 (33.9%) |
| 45–64 | 74,258 (28.6%) | 33,166 (15.9%) | 1,559,808 (23.6%) |
| 65–74 | 42,307 (16.3%) | 36,466 (17.5%) | 301,356 (4.6%) |
| 75–84 | 47,783 (18.4%) | 59,841 (28.7%) | 198,201 (3.0%) |
| 85+ | 35,663 (13.7%) | 58,601 (28.1%) | 155,880 (2.4%) |
| **Sex** | | | |
| Male | 135,419 (52.1%) | 108,518 (52.0%) | 3,151,711 (47.6%) |
| Female | 124,308 (47.9%) | 100,333 (48.0%) | 3,464,520 (52.4%) |
| **Ethnicity** | | | |
| White | 195,496 (75.3%) | 185,078 (88.6%) | 5,100,116 (77.1%) |
| Asian | 32,191 (12.4%) | 9930 (4.8%) | 740,194 (11.2%) |
| Black | 15,394 (5.9%) | 6376 (3.1%) | 278,773 (4.2%) |
| Mixed/Other/Unknown | 16,646 (6.4%) | 7467 (3.6%) | 497,148 (7.5%) |
| **Region of residence** | | | |
| London | 46,854 (18.0%) | 28,976 (13.9%) | 1,077,599 (16.3%) |
| East Midlands | 22,571 (8.7%) | 18,721 (9.0%) | 586,513 (8.9%) |
| East of England | 26,323 (10.1%) | 22,696 (10.9%) | 678,869 (10.3%) |
| North East | 15,131 (5.8%) | 9403 (4.5%) | 376,600 (5.7%) |
| North West | 43,017 (16.6%) | 42,130 (20.2%) | 1,056,012 (16.0%) |
| South East | 32,740 (12.6%) | 28,785 (13.8%) | 887,116 (13.4%) |
| South West | 17,391 (6.7%) | 12,854 (6.2%) | 496,411 (7.5%) |
| West Midlands | 29,808 (11.5%) | 24,513 (11.7%) | 733,250 (11.1%) |
| Yorkshire and Humber | 25,892 (10.0%) | 20,773 (9.9%) | 723,861 (10.9%) |
| **Index of multiple deprivation** | | | |
| 1st quintile (most deprived) | 73,100 (28.1%) | 51,176 (24.5%) | 1,564,464 (23.6%) |
| 2nd quintile | 59,754 (23.0%) | 44,886 (21.5%) | 1,440,534 (21.8%) |
| 3rd quintile | 48,458 (18.7%) | 40,595 (19.4%) | 1,287,972 (19.5%) |
| 4th quintile | 42,608 (16.4%) | 38,396 (18.4%) | 1,209,869 (18.3%) |
| 5th quintile (least deprived) | 35,807 (13.8%) | 33,798 (16.2%) | 1,113,392 (16.8%) |
| **Month of hospital admission** | | | |
| Mar-20 | 12,408 (4.8%) | 17,564 (8.4%) | 31,598 (0.5%) |
| Apr-20 | 25,867 (10.0%) | 29,525 (14.1%) | 111,629 (1.7%) |
| May-20 | 7475 (2.9%) | 10,170 (4.9%) | 66,563 (1.0%) |
| Jun-20 | 2698 (1.0%) | 3740 (1.8%) | 25,007 (0.4%) |
| Jul-20 | 1077 (0.4%) | 1232 (0.6%) | 18,905 (0.3%) |
| Aug-20 | 911 (0.4%) | 514 (0.2%) | 29,130 (0.4%) |
| Sep-20 | 3945 (1.5%) | 2092 (1.0%) | 124,294 (1.9%) |
| Oct-20 | 15,235 (5.9%) | 14,004 (6.7%) | 474,083 (7.2%) |
| Nov-20 | 23,218 (8.9%) | 22,905 (11.0%) | 518,220 (7.8%) |
| Dec-20 | 31,468 (12.1%) | 31,564 (15.1%) | 852,653 (12.9%) |
| Jan-21 | 60,389 (23.3%) | 40,122 (19.2%) | 1,068,457 (16.1%) |
| Feb-21 | 19,503 (7.5%) | 12,959 (6.2%) | 289,488 (4.4%) |

**Table 1 (continued) | Characteristics of the study population compared with all people hospital-onset COVID-19 in England and all people with PCR-confirmed community-acquired COVID-19 in England**

| Characteristic | Study population (hospitalised for COVID-19 in England) n (%) | All people with hospital-onset COVID-19 in England n (%) | All people with PCR-confirmed community-acquired COVID-19 in England n (%) |
|---|---|---|---|
| Mar-21 | 5809 (2.2%) | 3750 (1.8%) | 134,516 (2.0%) |
| Apr-21 | 1929 (0.7%) | 1110 (0.5%) | 58,209 (0.9%) |
| May-21 | 1370 (0.5%) | 477 (0.2%) | 61,016 (0.9%) |
| Jun-21 | 3756 (1.4%) | 1325 (0.6%) | 293,512 (4.4%) |
| Jul-21 | 13,984 (5.4%) | 4377 (2.1%) | 926,889 (14.0%) |
| Aug-21 | 15,578 (6.0%) | 6394 (3.1%) | 775,051 (11.7%) |
| Sep-21 | 13,107 (5.0%) | 5027 (2.4%) | 757,011 (11.4%) |
| **Hospital outcome** | | | |
| Death | 51,948 (20.0%) | 69,243 (33.2%) | N/A |
| Discharge | 191,663 (73.8%) | 107,815 (51.6%) | N/A |
| Right-censored in hospital | 16,116 (6.2%) | 31,793 (15.2%) | N/A |
| **Median length of stay following admission/positive test (days)** | | | |
| Death | 8 days | 11 days | N/A |
| Discharge | 5 days | 13 days | N/A |
| **Vaccination status at date of admission (for admissions occurring January and July 2021)** | | | |
| Unvaccinated | 97,441 (72.0%) | N/A | N/A |
| <21 days after first dose | 10,774 (8.0%) | N/A | N/A |
| ≥21 days after first dose | 7885 (5.8%) | N/A | N/A |
| ≥14 days after second dose | 19,325 (14.3%) | N/A | N/A |
| **Charlson comorbidity index** | | | |
| 0 | 92,753 (38.8%) | N/A | N/A |
| 1–2 | 93,436 (39.1%) | N/A | N/A |
| 3–4 | 35,527 (14.9%) | N/A | N/A |
| 5+ | 17,190 (7.2%) | N/A | N/A |
| **Hospital load at time of admission (as proportion of busiest week)** | | | |
| 0–20% | 61,406 (23.6%) | N/A | N/A |
| 20–40% | 64,722 (24.9%) | N/A | N/A |
| 40–60% | 49,529 (19.1%) | N/A | N/A |
| 60–80% | 40,385 (15.5%) | N/A | N/A |
| 80–90% | 20,228 (7.8%) | N/A | N/A |
| 90–100% | 23,457 (9.0%) | N/A | N/A |
| **Route of admission** | | | |
| Via emergency ward | 204,151 (78.6%) | N/A | N/A |
| Directly to hospital | 55,576 (21.4%) | N/A | N/A |

## Results

### Participant characteristics

Among 259,727 people with COVID-19 hospitalised between 15th March 2020 and 30th September 2021, a total of 51,948 (20.0%) died, 191,663 (73.8%) were discharged and the remaining 16,116 (6.2%) remained in hospital at the date of data extraction and/or were right-censored at 90 days (Table 1). Figure 1 presents weekly hospital admissions for COVID-19 over the study period, with an indication of the first, second, and third waves. Figure 2 shows the vaccination status of the study

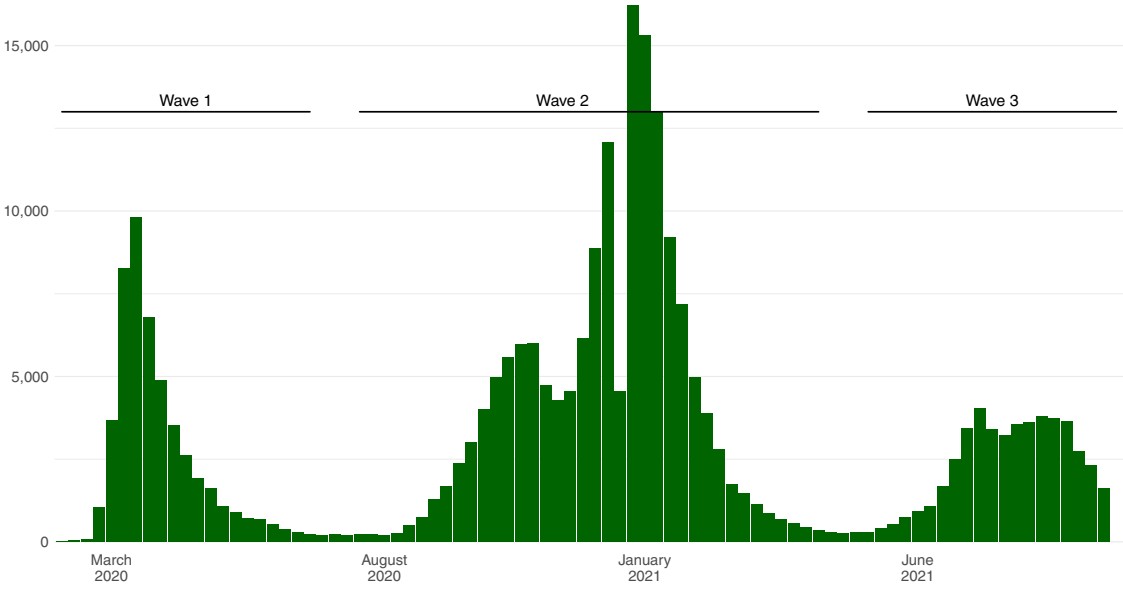

**Fig. 1 | Observed number of individuals hospitalised with COVID-19, by week of admission.** March 2020 to September 2021. Annotations shown for wave 1, wave 2, and wave 3. *n* = 259,727 individuals.

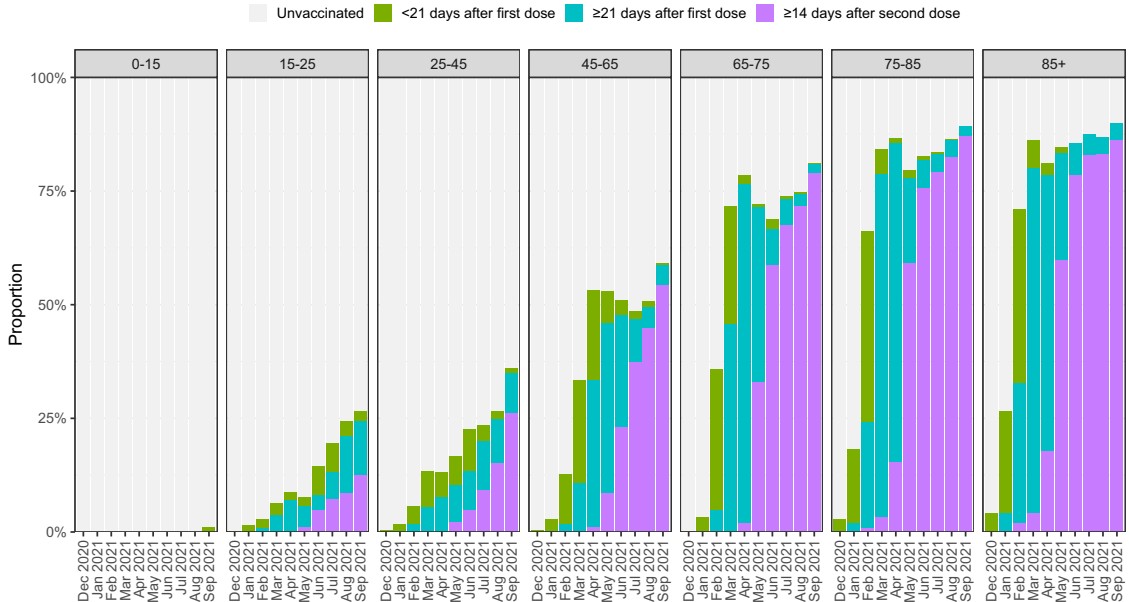

**Fig. 2 | Vaccination status of hospitalised individuals, by month of admission and age group.** December 2020 to September 2021. *n* = 166,893 individuals with vaccination status reported.

population between December 2020 and September 2021, by month of admission and age group, demonstrating the strong correlation between age and vaccination during the second and third waves.

Table 1 presents patient characteristics for the study population. Compared to all people with PCR-confirmed community-acquired COVID-19 infection, those hospitalised for COVID-19 were older (48.4% aged over 65 vs. 10.0%), more likely to be male (52.1% vs. 47.6%), and to reside in London (18.0% vs. 16.3%). A greater proportion of those hospitalised were of Black ethnicity (5.9% vs. 4.2%) and lived in an area of high deprivation (28.1% vs. 23.6%), compared to all those with community-acquired COVID-19. Comparative information on comorbidity was not available, although 7.2% of those hospitalised

had a Charlson comorbidity index (CCI) score of 5 or more (Table 1) compared to 2.3% among a sample of 657,264 individuals 20 years and older registered at English primary care practices in 2005[10].

## Observed hospital outcomes
In unadjusted comparisons, older individuals experienced poorer outcomes following hospital admission; almost half (46.5%) of those aged 85+ died in hospital compared to just 0.5% of those aged 15–24. Similarly, males (as compared to females), those living outside of London and the South West, and those with an increased CCI score (compared to lower CCI) were more likely to die in hospital (Supplementary Table 1).

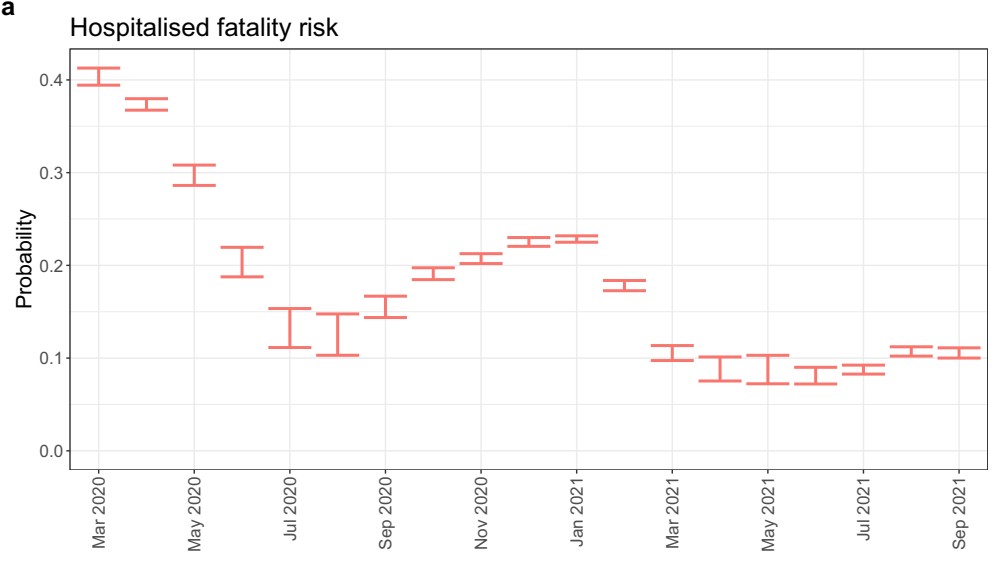

**a**

Hospitalised fatality risk

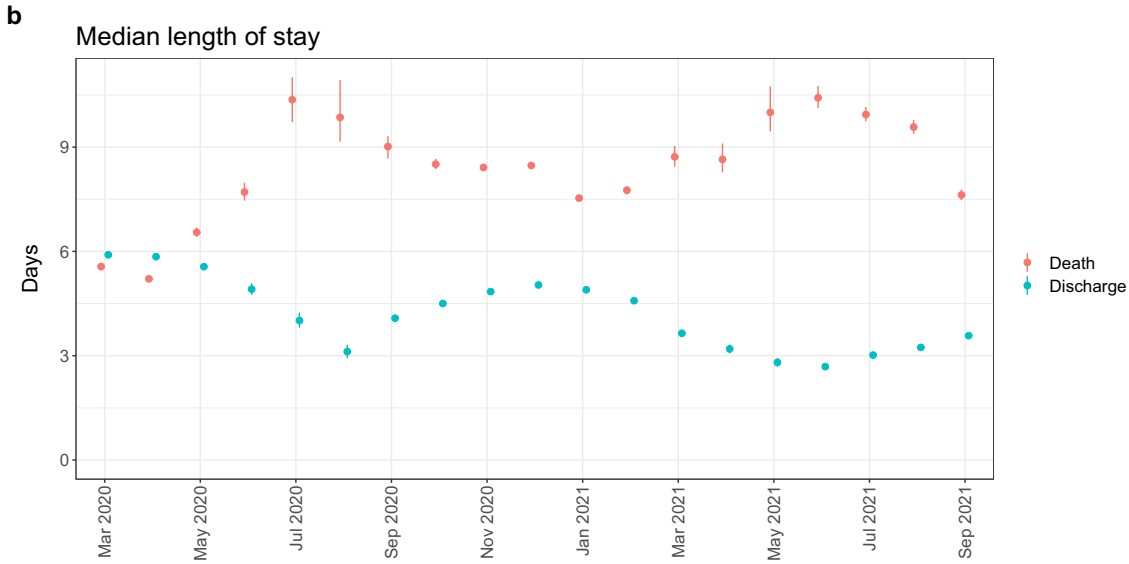

Error bars indicate 95% confidence intervals for HFR.

**b**

Median length of stay

Line ranges indicate 95% confidence intervals around median.

**Fig. 3 | Hospitalised fatality risk and median length of stay by month of hospital admission. a** Hospitalised fatality risk and **b** median length of stay in hospital prior to death or discharge by month of hospital admission. March 2020 to September 2021. Unadjusted for other covariates. $n$ = 259,727 individuals. Error bars are 95% confidence intervals.

## Hospitalised case-fatality risk

Figure 3a shows estimated HFR by month of hospital admission. HFR decreased during the first wave of the pandemic from 40.3% (95% confidence interval: 39.4–41.3%) in March 2020 to 12.3% (10.3–14.8%) in August 2020. During the second wave HFR increased to a peak of 22.8% (22.5–23.2%) in January 2021, although by March 2021 had halved to 10.5% (9.7–11.4%) and has remained at or below 10% throughout subsequent months (Supplementary Table 2).

In estimates of HFR by month and a single other covariate i.e. unadjusted for all other covariates older individuals had increased HFR. In March 2020 HFR was 1.4% (0.4–5.7%) among those aged 0–14, 22.3% (20.9–23.9%) for those aged 45–64 and 66.5% (64.4–68.7%) among those aged 85 + (Fig. 4a, Supplementary Table 3). Males tended to have greater HFR than females, although the extent of this differed by month, being more evident in March 2020 (HFR of 43.7% (42.5–44.9%) among males compared to 35.2% (33.8–36.6%) among females) than in April 2021 (HFR of 8.6% (6.9–10.6%) among males

compared to 8.9% (7.2–10.9%) among females) (Supplementary Fig. 1, Supplementary Table 4). HFR was lower for those residing in London and the South West: in December 2020 HFR was 18.7% (17.8–19.6%) in London and 17.5% (15.8–19.4%) in the South West, with a point estimate above 22% in all other regions of England (Supplementary Fig. 3, Supplementary Table 6). Lastly, HFR was increased among those with a higher CCI; in December 2020 HFR was 6.7% (6.2–7.2%) among those with a CCI of 0, 22.4% (21.6–23.1%) with a CCI of 1–2, 38.7% (37.3–40.2%) with a CCI of 3–4, and 47.3% (45.3–49.4%) with a CCI of 5 or above. This association was seen throughout the study period (Supplementary Fig. 4, Supplementary, Table 7).

During the initial months of the study those admitted to hospitals which were experiencing higher load had a greater HFR as compared to those admitted to hospitals with lower activity e.g. during March 2020, HFR was 44.9% (39.6–50.9%) for hospitals at 90–100% of their peak load, compared to 38.8% (36.6–41.0%) for hospitals at 0–20% of their peak load. This disparity appeared to lessen during the summer

**Table 2 | Hospitalised fatality risk by vaccine status at hospital admission and age group**

| Age group | Unvaccinated | <21 days after first dose | ≥21 days after first dose | ≥14 days after second dose |
|---|---|---|---|---|
| [0,15) | 0.2% (0.1–0.4%) | – | – | – |
| [15,25) | 0.3% (0.2–0.5%) | 1.3% (0.3–5.3%) | – | – |
| [25,45) | 1.7% (1.5–1.9%) | 2.1% (1.3–3.5%) | 1.2% (0.7–2.2%) | – |
| [45,65) | 10.1% (9.8–10.5%) | 8.7% (7.5–10.1%) | 6.1% (4.9–7.7%) | 7.5% (6.2–9.0%) |
| [65,75) | 25.3% (24.5–26.0%) | 22.5% (20.5–24.7%) | 18.2% (15.6–21.1%) | 14.9% (12.9–17.2%) |
| [75,85) | 38.6% (37.7–39.6%) | 34.5% (32.8–36.2%) | 26.2% (24.1–28.5%) | 22.5% (20.4–24.8%) |
| [85,Inf] | 49.9% (48.7–51.1%) | 47.0% (45.2–48.9%) | 37.7% (35.4–40.2%) | 32.0% (29.1–35.2%) |

Figures in brackets represent 95% confidence intervals. Estimates are replaced with a dash (–) where insufficient information was available.

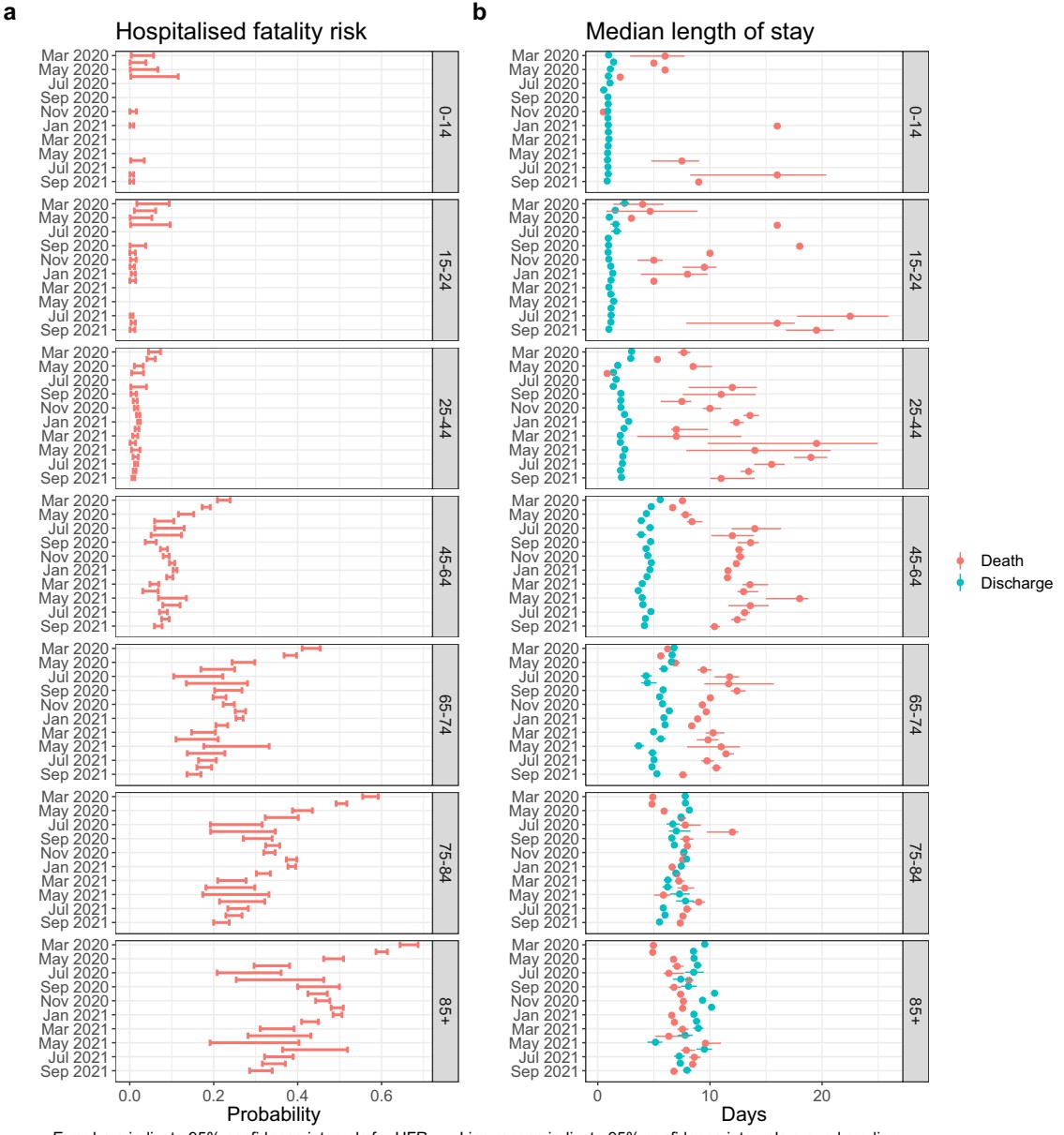

**Fig. 4 | Hospitalised fatality risk and median length of stay by month of hospital admission and age group. a** Hospitalised fatality risk and **b** median length of stay in hospital prior to death or discharge by month of hospital admission and age group. March 2020 to September 2021. Unadjusted for other covariates. n = 259,727 individuals. Error bars are 95% confidence intervals.

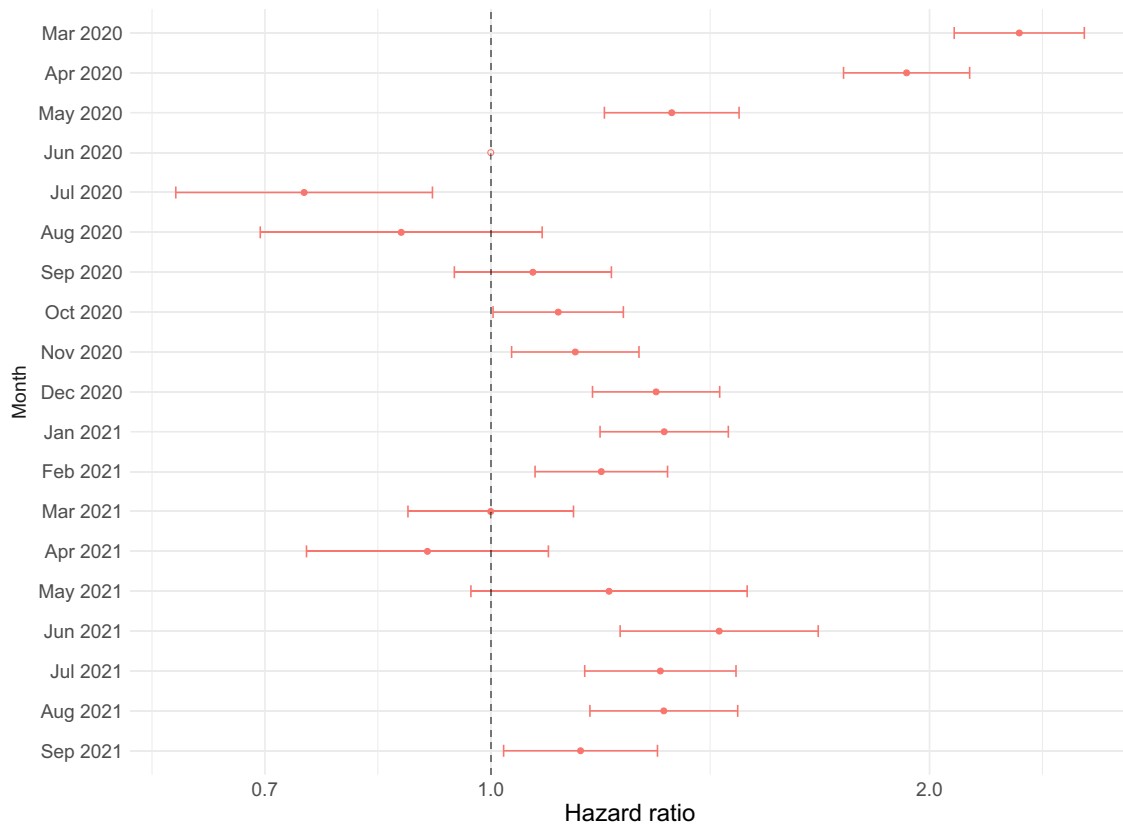

**Fig. 5 | Hospitalised fatality sub-distribution hazard ratio by month of hospital admission.** March 2020 to September 2021. Model includes stratification on age group, region of residence, and vaccination status, and regression adjustment (main effects) on month of hospital admission, sex, ethnicity, IMD quintile, hospital load, and CCI. Reference group: June 2020. n = 238,897 individuals with necessary information reported. Figure shows point estimate of hazard ratio with 95% confidence intervals.

months, but was seen again during autumn of 2020, as admissions began rising. In November 2020, HFR was 23.6% (21.1–26.4%) for hospitals at 90–100% of peak load, compared to 17.4% (16.4–18.6%) for hospitals at 0–20% (Supplementary Table 8).

### Length of stay
Figure 3b shows estimated median lengths of stay until death or discharge by month of hospital admission. Aside from the first 2 months of the pandemic (March–April 2020), those with an eventual outcome of death had longer stays in hospital compared to those who were discharged. The length of stay prior to death and discharge followed approximately inverse trends: whilst length of stay prior to discharge decreased throughout the first wave, from 5.9 (5.8–6.0) days in March 2020 to 3.1 (2.9–3.3) days in August 2020, length of stay prior to death increased from 5.6 (5.5–5.6) days to 9.9 (9.2–10.9) days.

During the second wave, lengths of stay prior to discharge initially lengthened, peaking at 5.0 (5.0–5.1) days in December 2020, before falling to 2.7 (2.6–2.8) days by June 2021. Conversely, length of stay prior to death was shortest in January, at 7.5 (7.5–7.6) days, and lengthened to 10.4 (10.1–10.8) days by June 2021 (Supplementary Table 2).

Examining the estimated lengths of stay for different subgroups (Fig. 4b, Supplementary Figs. 1–5, Supplementary Tables 3–8), similar patterns were observed for males and females, although with less pronounced variation among males. Length of stay prior to death estimates were imprecise for younger individuals, due to the small number of events, but for age groups 45–64 and above, the median length of stay prior to death decreased with increasing age. Meanwhile older individuals had longer stays in hospital prior to discharge compared to younger individuals; the median time to discharge among those aged 85+ ranged between 5.1 and 10.4 days, compared to between 0.8 and 2.4 days for those aged 0–14 and 15–24 (Fig. 4b). Those with a higher CCI similarly experienced shorter stays prior to death and longer stays prior to discharge; in December 2020 those with a CCI above 5 remained in hospital for a median of 7.0 (6.8–7.2) days prior to death and 8.9 (8.7–9.0) days prior to discharge, compared to 10.6 (10.3–11.0) days and 3.4 (3.3–3.4) days for those with a CCI of 0. Hospitals experiencing higher load had similar lengths of stay until discharge but shorter lengths of stay until death as compared to those with lower activity.

### HFR by vaccination status
Table 2 shows HFR by vaccination status and age group. For all ages, vaccination was associated with a reduced HFR, with significant reductions in HFR among those hospitalised ≥21 days post first vaccine or ≥14 days post second vaccine. The HFR for a double vaccinated adult aged 75–85 was 22.5% (20.4–24.8%), this compares to 38.6% (37.7–39.6%) for an unvaccinated adult in the same age group and 25.3% (24.5–26.0%) for an unvaccinated adult aged 65–75.

### Relative risks
Figure 5 presents hazard ratios for hospitalised fatality by month of admission. Controlling for age group, region of residence, vaccination status, sex, ethnicity, index of multiple deprivation (IMD) quintile, CCI and hospital load, month of hospital admission remained a significant factor for the prognosis of hospitalised individuals. Compared to June 2020, the hazard for hospitalised fatality was increased during March–May 2020, September 2020–February 2021, and June–September 2021 (Fig. 5, Supplementary Table 9).

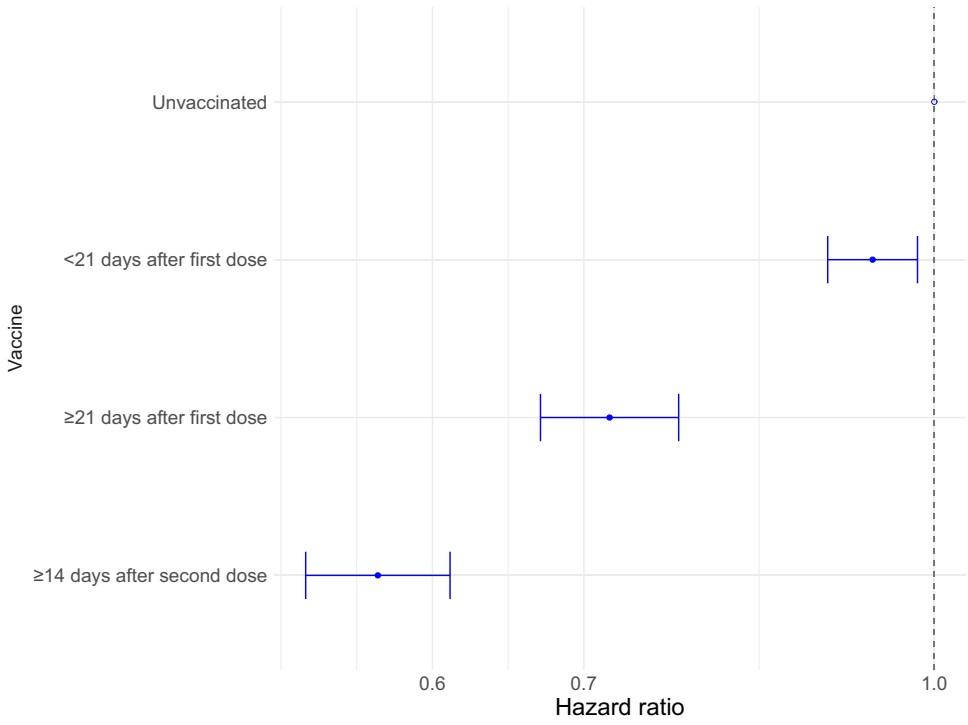

**Fig. 6 | Hospitalised fatality sub-distribution hazard ratio by vaccine status.**
January 2021 to September 2021. Model includes stratification on age group, region
of residence, and month of hospital admission, and regression adjustment (main
effects) on vaccine status, sex, ethnicity, IMD quintile, hospital load, and CCI.

Reference group: Unvaccinated. $n = 126,679$ individuals with necessary information
reported. Figure shows point estimate of hazard ratio with 95% confidence
intervals.

Similarly, controlling for month of admission and the factors
mentioned above, vaccination status was a significant factor for
prognosis following hospitalisation. During January–September 2021,
compared to the reference category of unvaccinated, the hazard ratio
for hospitalised fatality was 0.93 (0.89–0.98) for individuals hospita-
lised <21 days after first vaccination dose, 0.71 (0.67–0.77) for indivi-
duals hospitalised ≥21 days after first vaccination dose, and 0.56
(0.52–0.61) for individuals hospitalised ≥14 days after second vacci-
nation dose (Fig. 6, Supplementary Table 10).

There was an increased hazard of hospitalised fatality for those
of Asian ethnicity (1.19 (1.13–1.25)) but reduced for those of Black
ethnicity (0.90 (0.84–0.97)) compared to reference category
White ethnicity. Males had a greater hazard compared to females
(1.28 (1.24–1.32)), and compared to a CCI of 0, those with a higher
burden of comorbidity had a greater hazard of hospitalised fatality
(3.46 (3.27–3.67) for CCI of 5 and above). Those residing in more
deprived quintiles had greater hazards for hospitalised fatality
(1.10 (1.05–1.15) for the most deprived quintile) compared to a
reference of least deprived (Supplementary Fig. 6, Supplementary
Table 10).

Hazards were also elevated with increased hospital load, up to 1.23
(1.12–1.34) for load at 90–100% of the busiest week (compared to
0–20% load).

**Sensitivity to epidemic phase bias**
Supplementary Fig. 7 shows the outcome of the shift sensitivity ana-
lyses by month of symptom onset, adjusted for the same covariates as
above. The greatest effect was observed for the March 2020 hazard
ratio estimate, which steadily reduced towards 1 following a shift of
$c = 1, 2, 3$ or 4 days. The effect in other months was small, with the
previously described monthly trends persisting, although the slight
reduction in hazard estimated for the most recent month (September
2021) was no longer apparent.

## Discussion
People continue to experience hospitalisation for severe COVID-19, we
aimed to investigate how hospital prognosis and length of stay has
changed with the advent of vaccination and in the context of varying
hospital pressures. We examined absolute and relative risks of hospi-
talised fatality and lengths of stay in hospital during the first year and a
half of the COVID-19 pandemic in England.

### Findings in context
We identified a number of studies exploring COVID-19 HFR in England
according to demographic factors[1–3,7,11–13]. In line with these epide-
miological studies, we found that people with community-acquired
COVID-19 who became hospitalised were older, more likely to be male,
of Black ethnicity, and to live in areas of high deprivation, as compared
to everyone diagnosed with the virus. Among those who were hospi-
talised, we estimated greater absolute fatality risks for men and older
individuals, and HFR also varied according to ethnicity, month of
admission, hospital load, and region. Lengths of stay in hospital were
similarly associated with demographic factors, with median lengths of
stay prior to death typically longer than those prior to discharge. In
relative risk analyses controlling for all measured confounders, base-
line comorbidity burden was the strongest predictor of death.

Prior to this study there was limited information available on
COVID-19 outcomes at English hospitals by hospital load, although a
recent King's Fund study concluded that a shortage of overnight and
acute bed availability prior to the pandemic had already put hospitals
under increased strain[14]. In Switzerland meanwhile, increased hospital
load has been associated with poorer outcomes for COVID-19, with an
ICU occupancy of 70% or greater estimated to be a tipping point at
which outcomes became adversely affected[15].

Our estimates suggest a deterioration in survival as hospital load
increases, however, there are several potential biases which make this
finding hard to interpret. During periods of peak hospital load there is

likely a modification of an individual's willingness to attend healthcare services for mild illness, changes in admission criteria both to wards and intensive care units[16], and individuals with milder disease may be selected for transfer from overloaded hospitals to those with bed availability, due to the lower inherent risk. Each of these factors may influence the case-mix at times of peak hospital demand.

There is now compelling evidence that vaccination reduces the number of individuals being hospitalised[5] and the risk of mortality, regardless of hospital admission[9,17]. We found reduced hospitalised fatality among vaccinated individuals, with the reduction most clearly seen among older individuals. For those aged 75 and over, vaccination reduced HFR to approximately the risk of an unvaccinated individual aged 10 years younger. In adjusted estimates, each additional vaccine dose reduced the hazard for fatality by a significant margin, with a 42% (38–46%) estimated reduction in the risk of death for double-vaccinated individuals. This is a slightly lower reduction than for all community-acquired PCR-positive COVID-19 cases in England, where a 51% (37–62%) reduced risk of death was estimated for symptomatic individuals who had received a single vaccine[5]. This difference may reflect the portion of hospitalised individuals who die from other causes, or could be an indication of waning vaccine efficacy among our study population.

After controlling for all measured covariates, including hospital load and vaccination, we continued to estimate monthly variation in outcomes, with apparent seasonal variation in hazards. Whilst seasonal patterns in respiratory pathogens such as influenza and respiratory syncytial virus are well-documented[18,19], a multitude of interlinked factors including changes in national restrictions and the emergence of new variants may have influenced these trends.

### Strengths and limitations

The use of high-quality hospital surveillance data linked to several other comprehensive data sources is a strength of this study and enabled a broader understanding of the factors influencing hospitalised fatality. For covariates with varying levels of completeness we undertook sensitivity analyses to confirm minimal effects on our estimates (e.g. indication of injury as a factor for emergency care admission), however, there may have been other unmeasured confounders for which we could not account. Using appropriate statistical methods we adjusted for competing risks, and the use of a relatively coarse monthly timescale likely limited the extent to which our study was affected by epidemic phase bias[20]. Data linkage allowed for deaths occurring shortly after discharge to be identified, almost a fifth of all deaths in the study occurred within 14 days of discharge, suggesting palliative discharge.

Data on hospital pathways following admission were unavailable. As such, we were not able to subdivide the hospitalised population by severity of infection and/or need of respiratory support, whether within or outside of intensive care. Treatment data and changes in patient management were similarly unmeasured in our dataset, although the use of therapeutic agents is likely to have contributed to the reduction in hospital fatality risk, particularly at the start of the pandemic[7,21].

The measure of hospital burden we used considered acute hospital admissions at and around the time of admission as a proxy for bed occupancy. Whilst no single accepted measure of hospital burden exists, overnight bed occupancy is a widely used metric[14], and guidance on bed occupancy was issued to ICUs (e.g. alterations in practice upon reaching 150% and 200% above pre-pandemic baseline)[16]. A limitation of the bed occupancy measure is that it only measures demand and not supply (i.e. staffing levels), or the extent of other hospital pressures. Work to access and integrate measures of supply is ongoing.

Lastly, this study did not consider the significant proportion of individuals (up to 40%) who may have acquired COVID-19 nosocomially (in hospital). Fatality risks and lengths of hospital stay for these individuals are complicated by other conditions. Whilst not considered in our estimates, researchers in Scotland have found similar effects of age, sex, and comorbidity upon prognosis following nosocomial COVID-19 acquisition[22].

### Summary

Hospital outcomes and lengths of stay continue to vary according to case-mix, vaccination, and changes in hospital load more than 18 months after the pandemic began in England. One of the primary goals of the lockdown measures implemented in England at various times since the start of the pandemic has been to protect against hospitals becoming excessively overburdened. Even with these measures in place, being admitted during a period of high hospital load was correlated with poorer outcomes. Meanwhile, vaccinated individuals admitted to hospital for COVID-19 had a significantly reduced risk of mortality.

Outcomes following hospitalisation with COVID-19 should continue to be monitored, particularly with the emergence of new variants. The datasets and methods we describe continue to be vital to estimate changes in severity, providing an indication for demands on hospital resources, resulting effects on waiting lists for elective procedures, and monitoring the relationship between hospital burden and outcomes.

## Methods
### Study design and setting

A retrospective cohort study using competing risk regression to estimate relative risks of severe outcomes. We consider data on hospital admissions in England since the initial wave of COVID-19 in March 2020 until the end of September 2021, with follow-up until 22nd November 2021.

### Participants

All individuals aged 15 years and older with community-acquired COVID-19 (defined as a positive test for COVID-19 within −14 to 1 days of hospital admission), admitted to hospital in England for COVID-19 or another non-injury related condition between 1st March 2020 and 30th September 2021 were included ($n = 259,727$). Hospital records with inconsistent date information ($n = 2$) or missing demographic information ($n = 302$) were excluded.

### Data sources and outcomes

The United Kingdom Health Security Agency (UKHSA), alongside NHS England, monitors infectious diseases in England. The NHS England Secondary Uses Service (SUS) dataset contains well completed, accurate information on hospitalisation for COVID-19 in England, along with identifiers to augment these data through linkage to other routinely collected information. Admissions are, however, only entered into the SUS dataset upon completion of a hospital stay (i.e. at the point of discharge from hospital or death). So the information in SUS data was supplemented with information on individuals still in hospital though linkage to the Emergency Care Dataset for England (ECDS), which promptly records all emergency care attendances and onwards destinations (i.e. discharge home or admittance to hospital). Among those with completed hospital episodes, 77% were admitted via emergency care.

Complete information on deaths was obtained through linkage to the UKHSA deaths dataset, containing all dates of death for people with a positive COVID-19 test. Date of vaccination (first and second dose, third doses were not considered as only $n = 11$ hospitalised case had received a third dose during the follow-up period) was obtained through linkage to the UKHSA National Immunisation Management Service (NIMS). Testing information both for community-acquired COVID-19 infections identified through PCR testing on arrival at

hospital (Pillar 1) and PCR testing within the community (Pillar 2) were obtained from the UKHSA Second Generation Surveillance System (SGSS). All data were stored and analysed on UKHSA computers under agreed data governance protocols.

## Covariates

Covariates in the linked dataset included vaccination status (unvaccinated, <21 days of first dose, ≥21 days after first dose, ≥14 days after second dose), date of hospital admission (aggregated by month), age group, region of residence (Government office region), CCI[23], ethnicity, sex, IMD quintile, and measure of hospital load. The hospital load measure was defined as the number of COVID-19 admissions at an NHS trust within the 7 days around admission (3 before, same day, and 3 after), as a proportion of the busiest 7-day period at that trust. Hospital load was grouped into: 0–20%, 20–40%, 40–60%, 60–80%, 80–90%, and 90–100%. In relative risk analyses the two key exposure variables considered were vaccination status and month of hospital admission.

## Representativeness

Data comprised all new admissions for COVID-19 reported in England. Numbers of reported admissions were compared with the NHS weekly COVID-19 admissions data[24] to ensure data were representative. Hospital-onset COVID-19 (i.e. infection occurring in hospital) cases were excluded: those who acquired a hospital-onset infection during the study period ($n$ = 208,851) tended to be older and have longer lengths of stay than the community-onset cases considered in this study, with a greater proportion remaining in hospital post-90 days (Table 1). Hospital stays for these individuals may be influenced by conditions other than COVID-19, as described by ref. 25.

## Bias

Data validation was undertaken between the linked datasets, we found no systematic under-reporting or misreporting of person characteristics and linked information was used to minimise missing data. Censored outcomes and competing risks were explicitly accounted for by the choice of statistical method.

We carried out a sensitivity analysis to assess the potential effect of epidemic phase bias on the estimated hazard ratios in relative risk analyses[20]. This bias is caused by conditioning on an observed date later than the date of infection (see Supplementary Information for a full description), therefore in the sensitivity analysis we conditioned on date of symptom onset, which is nearer to date of infection than date of hospital admission. This conditioning ensured that the sensitivity analysis targeted bias due to epidemic phase, as opposed to any other factors which may influence time from symptom onset to admission.

## Statistical methods

In our study, hospitalised individuals are at risk of more than one event during the follow-up period i.e. they can die or be discharged. In this competing risk context standard survival analysis may result in biased estimates of the absolute and relative risks of hospitalised fatality, particularly when one of the competing risk is large (e.g. discharges to palliative care as a competing risk for death)[26]. Therefore, two alternative, complementary, statistical analyses were undertaken (see Supplementary Information for more details).

We used Aalen-Johansen cumulative incidence estimation to obtain estimates of cumulative HFR and median lengths of stay in hospital for specific sub-sets of the population, unadjusted for other factors[27]. Median lengths of stay were the weighted median estimate with weighted ties. We used stratified Fine-Grey competing risk regression with adjustment for confounders to estimate the association of each risk factor with the cumulative incidence of mortality within 90 days of hospitalisation with COVID-19 (we term this hospitalised fatality). Fine-Grey regression models the proportional sub-distribution hazard of hospitalised fatality derived from the

cumulative incidence function[28]. Stratification was used for confounders with non-proportional hazards (see Supplementary Information).

## Censoring

To focus our analyses on outcomes following COVID-19 admission, a pragmatic cut-off of 90 days from first positive specimen date was chosen and only those hospital outcomes (death or discharge) occurring within this cut-off were included. All records with outcomes occurring beyond 90 days ($n$ = 656) were right-censored at 90 days, while individuals who remained in hospital at the date of data extraction ($n$ = 15,460) were right-censored at the shorter of this date or 90 days. To better account for palliative discharge, deaths occurring within 14 days of discharge from hospital ($n$ = 9933, 19.1% of all deaths observed) were classified as deaths rather than discharges and the date of death used as the outcome date. Linkage to the UKHSA deaths data enabled these post-hospital discharge deaths to be identified.

## Model implementation

Statistical models were implemented using R version 4.1.1 (R Foundation for Statistical Computing, Vienna, Austria) and the open-source R packages survival v. 3.2–12[29], and matrixStats v. 0.60.0[30]. Figures were generated using the open-source R package ggplot2[31].

## Ethics approval

This study does not contain patient identifiable data. Consent from individuals involved in this study was not required. The mandatory surveillance systems used in this study, NHS England SUS, Emergency Care Dataset for England (ECDS), UKHSA deaths dataset, UKHSA NIMS, and UKHSA SGSS, are approved by the Department of Health and Social Care. Data were collected with permissions granted under Regulation 3 of The Health Service (Control of Patient Information) Regulations 2002, and without explicit patient permission under Section 251 of the NHS Act 2006.

## Patient and public involvement

This study was a retrospective cohort analysis. The research question, design and data collection were motivated by the response to an urgent public health emergency. The surveillance data were collected by NHS England and the UK Health Security Agency with permissions granted under Regulation 3 of The Health Service (Control of Patient Information) Regulations 2002, and without explicit patient permission under Section 251 of the NHS Act 2006. Although patients were not directly involved in the study design, the experiences of clinicians and public health officials interacting with patients informed the design of the data collection.

## Dissemination to participants and related patient and public communities

UKHSA and the MRC Biostatistics Unit have public facing websites and Twitter accounts @UKHSA and @MRC_BSU. UKHSA and the MRC Biostatistics Unit engage with print and internet press, television, radio, news and documentary programme makers.

## Transparency statement

The lead author affirms that this manuscript is an honest, accurate, and transparent account of the study being reported; that no important aspects of the study have been omitted; and that any discrepancies from the study as planned (and, if relevant, registered) have been explained.

## Reporting summary

Further information on research design is available in the Nature Research Reporting Summary linked to this article.

## Data availability

The data used in this study are protected data. These data are not publicly available because the information is personal or special category personal data, and there is risk of 're-identification' of data that has been anonymised by data matching, inference or deductive disclosure. Access to protected data is subject to robust governance protocols, where it is lawful, ethical and safe to do so. Individuals and organisations wishing to request access to data used in this study, from the NHS England Secondary Uses Service (SUS), Emergency Care Dataset for England (ECDS), UKHSA deaths dataset, UKHSA National Immunisation Management Service (NIMS), or UKHSA Second Generation Surveillance System (SGSS) can make a request directly to NHS Digital (https://digital.nhs.uk/services/data-access-request-service-dars) or to UKHSA (https://www.gov.uk/government/publications/accessing-ukhsa-protected-data). Access to protected data is always strictly controlled using legally binding data sharing contracts. Requests for underlying data cannot be granted by the authors because the data were acquired under licence/data sharing agreement from NHS Digital and UKHSA, for which conditions of use (and further use) apply. The estimated or observed values underlying each figure can be found in the Supplementary tables.

## Code availability

Code for the survival and matrixStats R packages is available from the Comprehensive R Archive Network (https://cran.r-project.org/). R code written to process the data, implement the statistical analysis, and produce the figures and tables is available online[32].

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

## Acknowledgements

We gratefully acknowledge all the clinicians, data reporters and individuals whose data were used in this study, as well as all UK Health Security Agency (UKHSA) colleagues involved in the COVID-19 response. We thank Shaun Seaman and Tommy Nyberg of the MRC Biostatistics Unit, Cambridge for their advice on epidemic phase bias, Kevin Fong and Tristan Caulfield at University College London Hospitals NHS Foundation Trust for discussion of the hospital load measure, James Stimson and members of the UKHSA Joint Modelling Team for the discussion and support of these analyses, and members of the UKHSA Immunisation Division for support with data preparation and linkage. This research is funded by the Medical Research Council (Unit programme number MC_UU_00002/11, PDK, DDA, AMP); a grant from the MRC UKRI/DHSC NIHR COVID-19 rapid response call (grant ref: MC_PC_19074, DDA, AMP); and the NIHR Health Protection Research Unit in Behavioural Science and Evaluation at University of Bristol, in partnership with UKHSA (D.D.A., A.M.P.). This research is funded by the Department of Health and Social Care using UK Aid Funding as part of the UK Vaccine Network, and is managed by NIHR (grant number PR-OD-1017-20006, D.D.A., A.M.P.). The views expressed in this publication are those of the author(s) and not necessarily those of the NIHR, the Department of Health and Social Care, or UKHSA. The funders had no influence on the methods, interpretation of results or decision to submit.

## Author contributions

P.D.K., P.B., A.M.P. and D.D.A. conceived the research study. A.C., R.H., S.E. and S.M. undertook data collection and dataset generation. P.D.K. and A.M.P. drafted the paper and formatted and verified the datasets. P.D.K. carried out the analyses. A.C., P.B., S.E., S.M., R.H. and D.D.A. provided expert advice and critical review of the paper prior to submission. The corresponding author attests that all listed authors meet authorship criteria and that no others meeting the criteria have been omitted.

## Competing interests

The authors declare no competing interests.
