## [Peer Review File · Nature Communications]

Trends in COVID-19 hospital outcomes in England before and after vaccine introduction, a cohort studyREVIEWER COMMENTS

Reviewer #1 (Remarks to the Author):

Paper review

Trends in hospitalised mortality risk and lengths of stay during the first, second and 2 current waves of COVID-19 in England: a cohort study

In this study the authors use data from the UK to estimate temporal trends in mortality and the impact of vaccination individuals hospitalised with approximately 260,000 individuals hospitalised with COVID-19.

This is an important study highlighting trends in COVID-19 mortality and the impact of vaccination and other sociodemographic characteristics in hospitalised individuals.

I believe this work will be of great interest to the readership of the journal.

General comments:

I would prefer that the authors refer to people included in the study as individuals or people throughout the manuscript rather than as patients.

Overall the rationale for the statistical approach in the background is clunky and could do with some editing. All of the points raised are important and correct, they just need to be made more clear for the generalist reader rather than a statistically minded audience. Some of the points raised would be better suited to the methods section, particularly around competing risks.

Specific comments:

Title: Could mention the impact of vaccination as that is a novel value added aspect of the study.

Background

Overall the paragraph starting at line 54 is difficult to follow and would benefit from some streamlining.

Line 56- this sentence seems incomplete? Would be easier to read to spell out the author/title of the example.

You then go on to discuss the issue of competing risks without first defining the term- an example of competing risks would be useful here. Not all people reading the paper may be familiar with this problem in the context of COVID-19.

Line 57 - the sentence

'If the competing risks of events are not independent, and the magnitude of the competing risk is large, the assumptions of conventional survival analysis methods such as Kaplan-Meier and Cox proportional hazards regression may provide biased estimates of risk, or of effect sizes on the rate of events, respectively'

Would benefit from being re-worded and broken up. It doesn't effectively convey what the authors mean. I am unclear what the phrase 'effect sizes on the rate of events' means.

Line 60- you say 'hospitalised mortality' would it be clearer to say 'mortality within 90 days of hospitalisation with COVID-19' This phrase also occurs in several other places in the manuscript.

Methods:

Line 123- you report data on representativeness which is very important and helps contextualise the study analysis. Is there any data to support the statement that those with hospital-onset infection were older had longer lengths of stay, which could be included in the supplementary material?

Censoring:

You included deaths within 14 days of a discharge as a death rather than a discharge. what proportion of mortality outcomes fell into this category? Did these differ by vaccination status?

Results:

I think the descriptive results of HFR by age, sex, ethnicity, and hospital load are of public health interest and would suggest moving these into the main text if there is space. Formal statistics comparing the HFR in subsequent waves against the initial wave as a reference would also be very interesting here though I appreciate this would add a significant amount of statistical analysis.

Impact of vaccination:

Before looking at the impact of vaccination on HFR it would be useful to briefly describe or comment on the characteristics of people with 0 vs 1 vs. 2 vaccinations. These people may be systematically different with respect to their hospitalisation and mortality risks and their profiles may change drastically over time as vaccination rollout began with the oldest and most vulnerable populations. This would greatly help the interpretation of the hazard ratios presented later in the results section.

Reviewer #2 (Remarks to the Author):

This is an ambitious paper that aims to measure hospitalised fatality rate in COVID patients, stratified in several different ways. It does some of these well and presents a great range of data, but the focus on so many ways of slicing the data I think makes the overall narrative of the paper less clear. Broadly, I think this paper could do with: a clearer rationale; some more care around the causal language used to interpret the results; and more discussion about what the results mean in practical terms, as detailed below.

Intro: My preference would be for a shorter, more punchy intro that sets out the rationale for the study. Some of the middle paragraphs of the intro might fit better in a "findings in context" type section in the discussion.

Hospitalised fatality risk: I think the dataset used limits the capability of this analysis to draw meaningful conclusions. As the authors elude to in the discussion, the case mix of the hospitalised population is likely to change dramatically during times of increased hospital load, with patients with less severe disease less likely to present to services (not wanting to "be a burden") or be admitted if they do present. Though there is adjustment for some patient factors, these will not capture important factors such as COVID severity. This should at the very least be discussed more thoroughly, and the authors should be careful to ensure that they don't imply they have shown that HFR is increased by busier hospitals.

Vaccination mortality risk: While the stratification of mortality risk by vaccination status is fine for a high level view, I don't think the analysis is sufficiently considered to demonstrate a causal link between vaccination and COVID mortality. While some basic adjustment is carried out, our own experience in this space has highlighted that the unvaccinated population is substantially (and increasingly) different to the vaccinated population. These differences are often not fully measurable in routine data such as this. The authors should therefore avoid language that implies causality throughout, two examples being: "the impact of vaccination among patients..." and "vaccinated 346 individuals admitted to hospital for COVID-19 had a significantly reduced risk of mortality, 347 and third (booster) doses may further reduce this risk". There may be other such examples.

"...is influenced by baseline demographic factors, vaccination status, and hospital load at admission." Is another example of causal language that should be avoided. "...varies by..." might be more appropriate.

Risk by demographics: These results are quite prominently highlighted in the abstract and discussion summary, but the results tables are in supplementary. Perhaps at least the headline demographics could be more prominently reported in the results?

Minor thing: Some of the tables 2, 4 and 5 are more or less entirely duplicating the results in the figures. While it's handy to have precise figures, these could perhaps be supplementary (I find the figure versions easier to comprehend at a glance).

Conclusions: While the authors describe to the variation in HFR they have identified, there is little discussion about what the practical implications of this variation might be. How might these findings affect healthcare services or research in future?

REVIEWER COMMENTS

Reviewer #1 (Remarks to the Author):

Paper review

Trends in hospitalised mortality risk and lengths of stay during the first, second and 2 current waves of COVID-19 in England: a cohort study

In this study the authors use data from the UK to estimate temporal trends in mortality and the impact of vaccination individuals hospitalised with approximately 260,000 individuals hospitalised with COVID-19.

This is an important study highlighting trends in COVID-19 mortality and the impact of vaccination and other sociodemographic characteristics in hospitalised individuals.

I believe this work will be of great interest to the readership of the journal.

Thank you for these comments and for providing this detailed review of our manuscript. We have provided a version of our manuscript with tracked changes which highlights the changes made in response to this review.

General comments:

I would prefer that the authors refer to people included in the study as individuals or people throughout the manuscript rather than as patients.

The term 'patient' was used to refer specifically to those admitted to hospital. However, we agree that the use of 'people admitted' or 'individuals admitted' to refer to those accessing healthcare services is generally more accepted and have made this change throughout when referring to individuals in the study.

Overall the rationale for the statistical approach in the background is clunky and could do with some editing. All of the points raised are important and correct, they just need to be made more clear for the generalist reader rather than a statistically minded audience. Some of the points raised would be better suited to the methods section, particularly around competing risks.

Thank you for the suggestion for a more focussed background section. We have moved the competing risks rationale and statistical details into the methods and supplementary information sections. We have also carefully edited the text to be clearer for a generalist reader.

Specific comments:

Title: Could mention the impact of vaccination as that is a novel value added aspect of the study.

We agree with this suggestion and have updated the title to 'Trends in COVID-19 hospital outcomes in England before and after vaccine introduction, 2020-2021: a cohort study'.

Background

Overall the paragraph starting at line 54 is difficult to follow and would benefit from some streamlining.

We have improved the description of survival analyses and moved the more statistical details into the methods and supplementary information sections.

Line 56- this sentence seems incomplete? Would be easier to read to spell out the author/title of the example.

Thank you for this suggestion, we have updated the reference to include the authors of this text.

You then go on to discuss the issue of competing risks without first defining the term- an example of competing risks would be useful here. Not all people reading the paper may be familiar with this problem in the context of COVID-19.

We have improved the description of competing risks and moved this rationale to the methods section.

Line 57 - the sentence

'If the competing risks of events are not independent, and the magnitude of the competing risk is large, the assumptions of conventional survival analysis methods such as Kaplan-Meier and Cox proportional hazards regression may provide biased estimates of risk, or of effect sizes on the rate of events, respectively'

Would benefit from being re-worded and broken up. It doesn't effectively convey what the authors mean. I am unclear what the phrase 'effect sizes on the rate of events' means.

Thank you for highlighting this section for improvement. We have updated the description of competing risks and moved this rationale to the methods section.

Line 60- you say 'hospitalised mortality' would it be clearer to say 'mortality within 90 days of hospitalisation with COVID-19' This phrase also occurs in several other places in the manuscript.

Thank you for this suggestion, we have altered the phrasing of this term in the background and given a definition of "hospitalised fatality" within the methods section, referred to as such thereafter.

Methods:

Line 123- you report data on representativeness which is very important and helps contextualise the study analysis. Is there any data to support the statement that those with hospital-onset infection were older had longer lengths of stay, which could be included in the supplementary material?

Whilst recently published and ongoing complementary work has explored hospital-onset infection during 2020 (Bhattacharya et al.) and 2021 (UKHSA, unpublished) we agree that the characteristics of hospital-onset individuals are usefully presented here to compare against those with community acquired COVID-19. We have now included a column in table 1 showing the demographic characteristics of these individuals.

Censoring:

You included deaths within 14 days of a discharge as a death rather than a discharge. what proportion of mortality outcomes fell into this category? Did these differ by vaccination status?

We have now included the number of outcomes in this post-discharge category within the methods; n=9,933, 19.1% of all deaths observed, and referred to this in the discussion. There was no difference in the proportion of deaths occurring pre or post-discharge according to vaccination status.

Results:

I think the descriptive results of HFR by age, sex, ethnicity, and hospital load are of public health interest and would suggest moving these into the main text if there is space. Formal statistics comparing the HFR in subsequent waves against the initial wave as a reference would also be very interesting here though I appreciate this would add a significant amount of statistical analysis.

We agree that selected results are of public health interest and have moved the HFR by age figures (figure 5) into the main results, as well as providing a fuller description of the changes in HFR by age, sex, and region. These descriptive results are unadjusted and hence not all results are described in detail.

The relative risks by month give an indication of significant differences over time demonstrated by the non-overlapping confidence intervals within the results section.

Impact of vaccination:

Before looking at the impact of vaccination on HFR it would be useful to briefly describe or comment on the characteristics of people with 0 vs 1 vs. 2 vaccinations. These people may be systematically different with respect to their hospitalisation and mortality risks and their profiles may change drastically over time as vaccination rollout began with the oldest and most vulnerable populations. This would greatly help the interpretation of the hazard ratios presented later in the results section.

Thank you for this suggestion, we have added a figure (figure 2) to show vaccine uptake in the hospitalised population over time, and a brief description within the results section. As highlighted by the reviewer, the vaccine rollout in England was initially only available to at-risk and older populations (before being steadily made available to younger age groups), and vaccination status is therefore strongly correlated with age group and comorbidity within our hospitalised cohort.

Reviewer #2 (Remarks to the Author):

This is an ambitious paper that aims to measure hospitalised fatality rate in COVID patients, stratified in several different ways. It does some of these well and presents a great range of data, but the focus on so many ways of slicing the data I think makes the overall narrative of the paper less clear. Broadly, I think this paper could do with: a clearer rationale; some more care around the causal language used to interpret the results; and more discussion about what the results mean in practical terms, as detailed below.

Thank you for these comments and for providing this detailed review of our manuscript. We have provided a version of our manuscript with tracked changes which highlights the changes made in response to this review.

Intro: My preference would be for a shorter, more punchy intro that sets out the rationale for the study. Some of the middle paragraphs of the intro might fit better in a “findings in context” type section in the discussion.

Thank you for the suggestion for a more focussed introductory section. As above, we have moved many of the statistical details to the methods and supplementary information sections. We have incorporated the paragraphs identified into a “findings in context” section within the discussion, expanding our comparison to other studies.

Hospitalised fatality risk: I think the dataset used limits the capability of this analysis to draw meaningful conclusions. As the authors elude to in the discussion, the case mix of the hospitalised population is likely to change dramatically during times of increased hospital load, with patients with less severe disease less likely to present to services (not wanting to “be a burden”) or be admitted if they do present. Though there is adjustment for some patient factors, these will not capture important factors such as COVID severity. This should at the very least be discussed more thoroughly, and the authors should be careful to ensure that they don’t imply they have shown that HFR is increased by busier hospitals.

Thank you for identifying this important point. We have included a more comprehensive discussion of the biases which affect the hospital load measure. As well as those presenting to services during times of peak infection being likely to have more severe disease than at times of reduced activity, selection criteria for patient transfer will also have impacted outcomes at hospitals experiencing additional load. Patients selected for transfer were those well enough to be transferred, with less severe symptoms and improved prognosis.

Vaccination mortality risk: While the stratification of mortality risk by vaccination status is fine for a high level view, I don’t think the analysis is sufficiently considered to demonstrate a causal link between vaccination and COVID mortality. While some basic adjustment is carried out, our own experience in this space has highlighted that the unvaccinated population is substantially (and increasingly) different to the vaccinated population. These differences are often not fully measurable in routine data such as this. The authors should therefore avoid language that implies causality throughout, two examples being: “the impact of vaccination among patients...” and “vaccinated 346 individuals admitted to hospital for COVID-19 had a significantly reduced risk of mortality, 347 and third (booster) doses may further reduce this risk”. There may be other such examples.

We agree with the reviewer's comment that the unvaccinated population are substantially and increasingly different to the vaccinated population and these differences would have begun to influence outcomes early on during the vaccine rollout. Without formal causal analysis we have taken care to reword our findings throughout the manuscript so as not to imply causality.

"..is influenced by baseline demographic factors, vaccination status, and hospital load at admission." Is another example of causal language that should be avoided. "...varies by..." might be more appropriate.

Thank you, as above we have taken care to reword our findings throughout the manuscript so as not to imply causality.

Risk by demographics: These results are quite prominently highlighted in the abstract and discussion summary, but the results tables are in supplementary. Perhaps at least the headline demographics could be more prominently reported in the results?

We agree that these results should be reported more prominently and have moved the HFR by age figures (figure 5) into the main results, and provided a fuller description of the changes in HFR by age, sex, ethnicity, comorbidity, and hospital load.

Minor thing: Some of the tables 2, 4 and 5 are more or less entirely duplicating the results in the figures. While it's handy to have precise figures, these could perhaps be supplementary (I find the figure versions easier to comprehend at a glance).

We agree that these tables are better suited to the supplementary information and have moved the results into this section.

Conclusions: While the authors describe to the variation in HFR they have identified, there is little discussion about what the practical implications of this variation might be. How might these findings affect healthcare services or research in future?

Thank you for this comment, we have expanded the conclusions section to provide more detail on how these findings are being applied to public health research and implications for ongoing research into the crossover between hospital demand and outcome.